# Use of metabolomics for predicting spontaneous preterm birth in asymptomatic pregnant women: protocol for a systematic review and meta-analysis

Renato T Souza,[1] Rafael Bessa Galvão,[1] Debora Farias Batista Leite,[2,3] Renato Passini Jr,[4] Philip Baker,[5] Jose Guilherme Cecatti[6]

For numbered affiliations see end of article.

**Correspondence to**
Professor Jose Guilherme Cecatti;
cecatti@unicamp.br

## ABSTRACT

**Introduction** Preterm birth (PTB) is the leading cause of neonatal mortality and short- and long-term morbidity. The aetiology and pathophysiology of spontaneous PTB (sPTB) are still unclear, which makes the identification of reliable and accurate predictor markers more difficult, particularly for unscreened or asymptomatic women. Metabolomics biomarkers have been demonstrated to be potentially accurate biomarkers for many disorders with complex mechanisms such as PTB. Therefore, we aim to perform a systematic review of metabolomics markers associated with sPTB. Our research question is 'What is the performance of metabolomics for predicting spontaneous preterm birth in asymptomatic pregnant women?'

**Methods and analysis** We will focus on studies assessing metabolomics techniques for predicting sPTB in asymptomatic pregnant women. We will conduct a comprehensive systematic review of the literature from the last 10 years. Only observational cohort and case-control studies will be included. Our search strategy will be carried out by two independent reviewers, who will scan title and abstract before carrying out a full review of the article. The scientific databases to be explored include PubMed, MedLine, ScieLo, EMBASE, LILACS, Web of Science, Scopus and others.

**Ethics and dissemination** This systematic review protocol does not require ethical approval. We intend to disseminate our findings in scientific peer-reviewed journal, the Preterm SAMBA study open access website, specialists' conferences and to our funding agencies.

**PROSPERO registration number** CRD42018100172.

## Strengths and limitations of this study

► This systematic review protocol takes into account some important aspects regarding conducting a systematic review about spontaneous preterm birth (sPTB) and metabolomics such as the criteria used for defining sPTB, different population risk stratification, method used to estimate gestational age and metabolomics techniques details.

► Two independent reviewers are responsible for searching and selecting studies, as also extracting data, and a third reviewer will resolve any disagreement.

► If possible, proper statistical methods will be applied to investigate metabolomics accuracy in predicting sPTB.

► Possible limitations to this review include the different criteria applied for defining sPTB, and the diverse population risk stratification.

## INTRODUCTION

Spontaneous preterm birth (sPTB) is the leading cause of perinatal mortality and short- and long-term morbidity.[1 2] It is defined as birth that occurs before 37 weeks gestation due to spontaneous onset of labour or preterm premature rupture of membranes (pPROM).[3 4] Several pathways and mechanisms linked with PTB have been proposed including, neuroendocrine, vascular, immune-inflammatory and behavioural processes.[5] More specifically, several markers associated with uterine distension/contraction, decidual inflammation/infection and activation of hypothalamic-pituitary-adrenal axis had been studied in the past decades.[5 6] However, no single marker or combination of markers has been found to be accurate enough for predicting sPTB.[7–10] History of previous PTB, cervical length at second trimester and cervico-vaginal fetal fibronectin (fFN) are the most promising clinical tests for predicting spontaneous preterm, but they seem not to be clinically useful for asymptomatic women. Sensitivity of short cervical length (<25 mm) and high cervico-vaginal fFN (>50 ng/mL) are around 33%–36% and 46%, respectively.[11–13]

PTB is a complex and multifactorial syndrome that possibly has a long pre-clinical

**BMJ**

phase, maternal and fetal interactions, genetic and environmental influences, and adaptive mechanisms.[14 15] These challenging aspects, and the presence of still unknown underlying mechanisms, are the main limitations for the identification of an accurate predictor for sPTB.[16–18] None of the predictors used in clinical practice, such as previous history of PTB, infection (vaginal and urinary contaminants), FN and transvaginal ultrasonography cervical length demonstrated exceptional accuracy for predicting sPTB.[7] An exploration of innovative approaches is urgently required.

Metabolomics is the study of metabolites, through identification and quantification of low-weight molecular particles, ie, tens to hundreds thousands of intermediate products and substrates of systems biology chemical reactions.[19 20] This novel approach has been applied for identifying biomarkers and underlying biochemical pathways associated with complex obstetrical syndromes as pre-eclampsia, fetal growth restriction, gestational diabetes and PTB. In contrast to other 'Omics Sciences' techniques, metabolomics is more closely associated with the phenotype of the disease and might thus identify a more robust and reliable set of predictors.[21] Importantly, implementing an adequate Omics experimental design is crucial for metabolomics studies. Using different baseline population (asymptomatic vs symptomatic or low- vs high-risk women for developing sPTB), study designs (prospective cohorts, case–control or cross sectional studies), sources of samples (amniotic fluid, vaginal fluid, blood, urine, hair, etc) and the timing of sample collection each have significant effects on study findings and the consequent interpretation and contribution to the current gap of knowledge.[19]

Different reviews collating scientific knowledge regarding PTB biomarkers/predictors has been conducted. Different methodology approaches has been applied so far, including narrative, systematic and umbrella reviews, a more comprehensive review that includes not only original studies but also other reviews.[7 22–24] At the best of our knowledge, there is no systematic review on metabolomics markers. Therefore, we aim to conduct a systematic review of original studies investigating the use of metabolomics biomarkers for predicting sPTB in asymptomatic pregnant women. This protocol describes the methods that will be applied in our systematic review.

## METHODS AND ANALYSIS
The current systematic review proposal will be conducted, written and published following the Preferred Reporting Items for Systematics Reviews and Meta-Analyses (PRISMA-P) recommendations.[25]

### Review question
What is the performance of metabolomics for predicting sPTB in asymptomatic pregnant women?

### Eligibility criteria
Original cohort or case–control studies involving asymptomatic pregnant women at the moment of sample collection (exposure) and with samples analysed using metabolomics techniques. Studies will be excluded if (1) they are cross-sectional studies, clinical trials, editorials, letter to editors, case reports, expert opinions, commentaries or any type of review; (2) they describe only experimental studies with animals; or (3) they are duplicated data (eg, data published in conferences proceedings and, then, published again in scientific journals). In this case, only the most complete publication will be considered, after comparing and confirming that the same technique and metabolites were explored. Studies published from 2008 to 2018 will be considered, and there will be no language restriction. Before submitting this systematic review for publication, we will rerun the search strategy to identify new studies that have been published after performing the first round of search.

### Participants
The current review is interested in evaluating the performance of metabolomics biomarkers for sPTB in asymptomatic pregnant women, which may contribute to clinical practice, potentially providing information regarding onset of preterm labour. Nevertheless, we aim to identify studies addressing only early predictors collected from women who are in an early preclinical stage, which might contribute to a wider window of opportunity for interventions and also to develop a widely reproducible screening test. Asymptomatic pregnant women should not have regular uterine tightening/contractions or signs of rupture of membranes (ie, watery discharge). In addition, the study should preferably have a standardised definition of sPTB, the outcome of interest.

### Information sources
The search will be held in the following databases: PubMed, EMBASE, Scopus, CINAHL, and Web of Science, BVS/BIREME, which includes the Latin American and Caribbean Health Sciences Literature (LILACS), Medline and the Scientific Electronic Library Online (Scielo). In addition, secondary sources of original studies will be explored such as Google Scholar, hand-held searching of the reference list of eligible studies, conference proceedings, and contact with authors when necessary.

### Search strategy
The following terms will be used in our search strategy for the different scientific databases: (preterm birth, premature birth, premature infant, premature labour, extremely premature infant, premature obstetric labour, spontaneous preterm birth, extreme preterm birth, late preterm birth, moderate preterm birth, preterm premature rupture of membranes, preterm delivery, PROM, sPTB, preterm PROM, pPROM, p-PROM) AND (metabolomic*, metabonomic*, metabolit*, lipidomic*, H NMR, proton NMR, proton nuclear magnetic resonance, liquid chromatogra*, gas chromatogra*, UPLC, ultra-performance liquid chromatograph*, ultra-performance liquid chromatograph*, HPLC, high performance liquid

chrromatograph*, high-performance liquid chrromatograph*) AND (pregnan*, antenat*, ante nat*, prenat*, pre nat*) (online supplementary material). Respective adaptations in the syntax of search for each database will be applied accordingly. No filters—such as 'research in animal's models' and 'reviews'—will be used in our search strategy, as it will be excluded according to eligibility criteria. The complete search strategy, including Boolean terms, is provided as online supplementary material.

### Data management

We will export search results to a reference manager (Mendeley). Then, the following information will be collected from each study using an appropriate form, which will be entered in an Excel spreadsheet: author's name, year of publication, country, study design, number of participants with and without sPTB, type of metabolomics analysis technique (liquid or gas chromatography, nuclear resonance), laboratory methods for metabolites data acquiring (targeted or untargeted techniques, etc), subtype of PTB (spontaneous preterm labour or pPROM), number of fetuses (singleton vs multiple), gestational age when samples were collected, source of samples (type/site of tissue), low- or high-risk for PTB (authors criteria used to define the population will be collected) and method applied to estimate gestational age. If possible, additional variables related to sPTB categories (delivery before 28 weeks and before 34 weeks) will be recorded for secondary analyses. Original authors will be contacted to clarify data, when needed. Finally, we will check the biochemical class of identified metabolites in Human Metabolome Database (HMDB, version 4.0) to explore and synthetize whether there are common biological pathways associated with sPTB.[20]

### Selection process

Two independent reviewers (RTS and RBFG) will be responsible for screening and selecting studies initially according to title or abstract. Both researchers will read the full text of non-excluded studies to discriminate eligibility. A third reviewer (DFBL) will consider any disagreement; additional reviewers (RPJ, PNB and JGC) will be responsible for supervising all steps and approving data extraction.

### Data collection process

We will extract search results to a reference manager where all studies will be stored. Then, included studies will be placed in a new folder. Finally, we will manually extract data of interest from these included studies to an Excel file. Each reviewer will have their own reference manager account, file and folder and discrepant results will be discussed together with the third reviewer.

### Outcomes and prioritisation

The primary outcome is sPTB, defined as any birth occurred before 37 weeks of gestation due to spontaneous onset of labour or pPROM. Secondary outcomes are:
1. sPTB before 28 weeks;

2. sPTB before 32 weeks;
3. sPTB before 34 weeks.

The capacity to predict different degrees of sPTB (categories of gestational age) is important as the extreme (<28 weeks), moderate (<32 weeks) and non-late preterm (<34 weeks) newborns have different adverse outcomes compared with non-extreme (≥28 weeks); non-moderate (≥32 weeks) or late (≥34 weeks) preterm newborns.

Ideally, the method of gestational age estimation should be clearly reported. For instance, it can be reported as estimated by last menstrual period (LMP) and confirmed by an early ultrasound or only by an early ultrasound when LMP is unknown/uncertain.

### Index test

Metabolomics techniques to predict sPTB is the diagnostic test of interest. Metabolomics is a technique to identify and quantify metabolites from biological samples using different type of platforms/equipment. The most common platforms include gas, liquid chromatography or ultra-performance liquid chromatography coupled to a mass spectrometer or a proton nuclear magnetic resonance.[26] The performance of the different thresholds of each metabolite will be compared and summarised through hierarchical summary receiver operator characteristic curve (meta-analysis) according to the subgroups described above. Considering that the raw data is not available in the majority of the diagnostic test accuracy studies[27] and that metabolites levels are usually reported as continuous variables, we intend to use a meta-analysis model based on receiver operating characteristic (ROC) curves.[28] Briefly, a two-parameter model, based on the estimation of α and β parameters (using standard errors or maximum likelihood), will be applied as reported by Kester and Buntinx.[28] Therefore, pooled ROC curves can be converted to a estimated ROC curve with 95% CI. This method can also be applied in categorical-ordinal variables tests.

### Risk of bias in individual studies

We will apply the Quality Assessment of Diagnostic Accuracy Studies (QUADAS-2) tool[24] to assess the risk of bias and applicability of primary diagnostic accuracy studies. Each study will be classified as 'low', 'high' or 'unclear' regarding risk of bias for each of the four domains of QUADAS tool: Patient Selection, Index Test (metabolomics), Reference Standard (occurrence of PTB) and Flow and Timing of participant's inclusion and follow-up. For example, studies will be labelled as 'low' risk of bias for Reference Standard when definition of sPTB and gestational age estimation are clear; 'high' risk of bias would be considered when the moment of sample collection is not well described.

### Data synthesis

We will report details of identification, screening, eligibility and included studies using a flow diagram, according to PRISMA recommendations.[25] Data from included studies

will be synthesised into tables according to the variables of interest. If possible, we will present data meta-analysis according to study design, metabolomics technique and type of samples analysed. We intend to perform subgroup analysis according to:

▶ different metabolomics methods applied: gas or liquid chromatography coupled with mass spectrometry or proton nuclear magnetic resonance;
▶ singleton and multiple pregnancies;
▶ low-risk and high-risk women for developing PTB;
▶ subtype of PTB: sPTB exclusively due to spontaneous onset of labour with intact membranes or sPTB due to PROM;
▶ gestational age interval when samples were collected: first trimester, second trimester and third trimester.

Heterogeneity will be assessed by Cochran's Q, Hotelling's T-squared ($\tau^2$) and $I^2$ tests. Funnel plots and sensitivity and cumulative analyses will be applied for detection of temporal trends and publication bias.

### Potential anticipated limitations to this review

First, although we have not considered any language restriction, we consider that there might be a limitation in studies published entirely in non-English language. However, in the last decade, more than 95% of scientific biomedical literature has been published in English,[29] then we consider this a minor selection bias. Second, we intend to stratify the groups according to population risk. However, the characterisation of low- or high-risk for sPTB is controversial and lacks standardisation, which might limit data comparison and subgroup analysis. Finally, categorisation of sPTB into spontaneous onset of labour or pPROM is another topic of potential limitation—the recognition of the main initial mechanism for preterm delivery might not always be possible. Even when specified, it might provoke uncertainty and could limit further considerations regarding preterm phenotypes. In addition, another limitation is that individual patient data will not be collected.

### Patient and public involvement

Patients will not be directly involved in the study and no experience or direct impact from their perspective can be discussed.

### ETHICS AND DISSEMINATION

We intend to disseminate our findings in scientific peer-reviewed journal, general free access website of Preterm Screening and Metabolomics in Brazil and Auckland (Preterm SAMBA) study, specialists' conferences, and to our funding agencies.

### DISCUSSION

This systematic review will comprise current knowledge related with metabolomics in the context of PTB prediction. Metabolomics science, a resourceful innovative field that allows better understanding on pathophysiology of complex syndromes, may address the main compounds associated with the spontaneous preterm delivery and, therefore, motivate further researchers to validate early measurable predictors of PTB.

Metabolomics performance for predicting sPTB remains unclear and standardised and high-quality studies are needed to clarify the clinical application of metabolites for predicting sPTB. Nevertheless, metabolomics discovery studies commonly requires further validation studies; reproducible methodology is crucial. This systematic review protocol will collate the main potential early biomarkers, subgroup analysis and standardised definition for sPTB to better understand metabolomics performance in predicting sPTB and also to show its heterogeneity in terms of methodology (samples used, metabolomics technique, definition of SPTB phenotype, etc). High performing predictors of PTB will help combat this leading cause of neonatal mortality and morbidity.

**Author affiliations**
[1]Obstetrics and Gynecology, Universidade Estadual de Campinas, Campinas, Brazil
[2]Department of Tocogynecology, Campinas' State University, Campinas, Brazil
[3]Department of Maternal and Infant Health, Universidade Federal de Pernambuco, Recife, Brazil
[4]Universidade Estadual de Campinas Faculdade de Ciencias Medicas, Campinas, Brazil
[5]University of Leicester, College of Medicine, Leicester, UK
[6]Obstetrics and Gynecology, University of Campinas, Campinas, Brazil

**Acknowledgements** Ana Paula de Morais e Oliveira, librarian of University of Campinas—Unicamp, Brazil, for collaborating in developing search strategy and Rachel Hanisch for her suggestions to some sections of the paper.

**Contributors** RTS and RBG conducted the systematic review as independent first reviewers. JGC, RP and DFBL made the conflicting decisions regarding papers selections. PB, RP and JGC participated in the systematic review conception, methodology and framework, together with all the others co-authors.

**Funding** This research was supported by Brazilian National Research Council (grant number 401636/2013-5) and Bill and Melinda Gates Foundation (grant number OPP1107597—Grand Challenges Brazil: Reducing the burden of preterm birth, FIOTEC number 05/2013), which provided funding to PRETERM-SAMBA project (www.medscinet.com/samba). RTS and DFBL have been awarded PhD scholarships from the CAPES Foundation, an agency under the Ministry of Education of Brazil, process 88881.134095/2016-01 and 8881.134512/2016-01, respectively.

**Competing interests** All authors are carrying original research about metabolomics and presenting conferences about this topic, including spontaneous preterm birth, pre-eclampsia, gestational diabetes mellitus and fetal growth restriction. Philip N Baker is principal investigator of Metabolomics Diagnostics Ltd, a company dedicated to develop innovative screening tests using metabolomics technology.

**Patient consent for publication** Not required.

**Ethics approval** This systematic review does not require ethical approval from the Research Council or Ethics board.

**Provenance and peer review** Not commissioned; externally peer reviewed.

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
