## [Reviewer comments · BMJ Open]

This paper was submitted to a another journal from BMJ but declined for publication following peer review. The authors addressed the reviewers' comments and submitted the revised paper to BMJ Open. The paper was subsequently accepted for publication at BMJ Open.

(This paper received three reviews from its previous journal but only two reviewers agreed to published their review.)

ARTICLE DETAILS

TITLE (PROVISIONAL)	The use of metabolomics for predicting spontaneous preterm birth in asymptomatic pregnant women: protocol for a systematic review and meta-analysis
AUTHORS	Souza, Renato T; Galvão, Rafael Bessa; Leite, Debora Farias Batista; Passini Jr, Renato; Baker, Philip; Cecatti, Jose Guilherme

VERSION 1 – REVIEW

REVIEWER	Giuseppe Rizzo Università di Roma Tor Vergata Division of Maternal Fetal Medicine Ospedale Cristo Re Rome Italy
REVIEW RETURNED	04-Sep-2018

GENERAL COMMENTS	To investigate the role of metabolic in predicting preterm birth is of clinical interest and i would like to congratulate with Authors for their effort in planning a systematic review and meta-analysis My suggestions are as follows 1)to consider as a secondary outcome also 32 weeks for PTB 2)to differentiate between spontaneous PTB and preterm premature of membranes (pPROM) 3)to consider in data analysis the "Umbrella review" methodology. (Lucaroni et al 2018 Biomarkers for predicting spontaneous preterm birth: an umbrella systematic review, The Journal of Maternal-Fetal & Neonatal Medicine, 31:6, 726-734) This paper can be added in the introduction
---

REVIEWER	Tormod Rogne Norwegian University of Science and Technology, Norway. Yale University, USA.
REVIEW RETURNED	19-Oct-2018

GENERAL COMMENTS	I read this protocol with great interest. I applaud all efforts to publish protocols as well as the willingness to systematically review the literature! Please find my comments below. They may seem a bit blunt, but it's only meant to be helpful. The section regarding analyses has the most room for improvement. All in all, this protocol creates a great fundament for an excellent review. The strengths and limitations sections following the keywords seems misplaced, and does not discuss limitations.
--

Introduction

It would be helpful with a more detailed description of what factors as of today (either alone or in combination) offer the best predictive value, and provide some estimate of this prediction.

Missing reference on sentence ending on line 84.

Please elaborate in a couple of more sentences on metabolomics vs other omics, and what is meant by "omics".

Are there any much-cited/controversial references on metabolomics in relation to PTB that inspired the researchers to dig into this topic?

Methods

Selection criteria: Case-control studies are unlikely to give representative results of diagnostic accuracy, so consider limiting to cohort and cross-sectional studies.

Participants: Please clarify whether pre-conceptional women are eligible (I assume not).

Search: The search seems to be good, but I would like to see the actual search that will be applied for each database. This search should include a mix of keywords and MeSH-terms (NB be aware of empty spaces). One keyword I can see missing is metabolom*

Data management: Although Mendeley is a very good reference manager, I would argue that EndNote is more efficient when you import hundreds of references from multiple databases, and then go through duplicates. Just as a suggestion.

Selection process: Seems to be very good, but will there be one or two reviewers to read the full-texts?

Data synthesis: I think it would be helpful with more pre-specified sub-analyses. One particular that comes to mind is sub-analysis based on when in pregnancy the test was done. I also suggest excluding twins from primary analyses.

The researchers should also describe how they will handle heterogeneity between studies (i.e. high I²). Important: It is strongly advised to refrain from doing meta-analyses if the individual studies are too clinically or statistically heterogeneous!

The researchers should clearly state all planned main analyses and sub-analyses. This includes how they are going to do the analyses. Which software to use. Are they using standardized means or means on original scale? Random effects or fixed effects? Adjustment for confounding? In addition to the mentioned ROC, is the main interest to present sensitivity and specificity, or present mean differences of metabolites between pregnant women that end up delivering preterm or non-preterm? How do they deal with missing data and publication bias (e.g. funnel plots)?

Potential limitations: I do not see why publication in non-English in itself should be a limitation. A very important limitation of this review is that the researchers have not considered collecting individual level data, nor considered asking the authors of the original studies to re-do some main analyses so that the analyses

	are conducted in the same way across studies. Heterogeneity between studies will likely be a big problem, as is often the case of SR of observational studies. Ethics: Could use criteria that only considering studies with stated approved by local research ethics committee.
--	--

VERSION 1 – AUTHOR RESPONSE

Reviewer 1

We acknowledge Prof. Giuseppe Rizzo concerns about the different phenotypes and potential clinical impact of distinct gestational ages of preterm birth or type of onset of labour. Firstly, we included now spontaneous preterm birth before 32 weeks as a secondary outcome and we will conduct analysis accordingly if data available for these outcomes allow (Page 9, line 194). Secondly, we have planned a secondary analysis to differentiate the spontaneous preterm birth from the preterm premature rupture of membranes (pPROM) (Page 11, line 230). Finally, we appreciate the suggestion of conducting an Umbrella Review, a broad approach that also includes already performed reviews, but authors decided to include only original studies in this systematic review because this is what had already been accorded in the original protocol assessed by peers during budget request for the Bill and Melinda Gates Foundation. Nevertheless, we acknowledge umbrella approach in the introduction session, informing to the audience the existence of alternative reviewing approaches (Lines 99-104, Page 5). We believe that our planned search strategy will cover published literature accordingly.

Reviewer 2

We also appreciate Prof. Tormod Rogne interest and constructive suggestions. As advised in the Submission Guidelines, we should state a Strengths and Limitations section after the Abstract. We have added a limitation as the fourth bullet point (Page 3, lines 59 and 60).

In the Introduction section, we added a piece of information regarding the current studied markers for predicting preterm birth and details of their performance (Page 4, Lines 71-75).

Regarding the Methods, we understand that majority of the diagnostic accuracy studies have a cross-sectional design. However, and in accordance with the Cochrane Handbook for Diagnostic Test Accuracy Reviews, cohort and case-control studies can also be considered. It is important to highlight that we intend to identify predictive biomarkers in asymptomatic pregnant women (as stated in Page 6, line 114-115), then studies assessing risk in pre-conceptional women are not eligible.

We comprehend the importance of reference managers, and appreciate the suggestion for using EndNote®. However, the Mendeley® is more familiar to the research team, and we hypothesize

that we would have the same performance with this tool. Furthermore, we have tried previously different literature search strategies, which are presented as Supplementary Material (Page 8, line 160); the term “metabolom*” has not changed the research results. In addition, we decided to use “free terms” instead of MeSH-Terms in order to standardize our search for all the different databases. We made clearer that two independent researchers will be involved in each step of literature search and study selection (Page 8, line 179-180), and we will contact the authors of the included studies if any clarification of data is necessary (Page 7, line 145 and Page 8, line 173).

We will collect data regarding the time of sample collection and data analysis will be performed considering this information – we will not compare tests performed at different gestational periods. We intend to analyse singleton and multiple pregnancies in separate, as described on Page 11, line 230-232.

Metabolomics techniques are heterogeneous and complement each other. Then, we speculate not to be possible to perform a quantitative analysis of individual metabolites. However, depending on data availability, we will perform a diagnostic accuracy meta-analysis.

We agree that a limitation of this study is that it will not collect individual patient data. We clarified that in line 247, Page 11.

Heterogeneity and publication bias will be addressed by proper statistical methods as described in lines 233-235, Page 11.

Finally, we are committed to change and update PROSPERO registration of this systematic review protocol accordingly.

VERSION 2 – REVIEW

REVIEWER	Giuseppe Rizzo Università Roma Tor Vergata Dept Maternal Fetal Medicine Ospedale Cristo Re Roma Italy
REVIEW RETURNED	04-Dec-2018
GENERAL COMMENTS	nicely reviewed
REVIEWER	Tormod Rogne Norwegian University of Science and Technology, Norway. Yale University, USA.
REVIEW RETURNED	21-Nov-2018
GENERAL COMMENTS	As before, a good protocol. Some of my previous suggestions have not been taken into account. I assume that the authors have elaborated to the editor why the suggestions have not been followed. I would nevertheless restate some aspects that I think

	will improve the systematic review:  - Please provide accurate search term, and include metabolom* as keyword. - Sub-analyses stratified by when in pregnancy the metabolomic test was carried out. - How the analyses will be carried out should be specified in more detail. And what do they do if there is considerable heterogeneity (both clinical and statistical)? In addition:  - Some of the newly inserted text has grammatical errors.
--	--

VERSION 2 – AUTHOR RESPONSE

Comments from Reviewer 1

- *Please provide accurate search term, and include metabolom* as keyword.*

There are a list of our search terms in the supplemental material and we added metabolome as a keyword (Line 48, Page 2).

- *Sub-analyses stratified by when in pregnancy the metabolomic test was carried out.*

We added a subgroup analysis according to gestational age when samples were collected (Lines 232-233, Page 11).

- *How the analyses will be carried out should be specified in more detail. And what do they do if there is considerable heterogeneity (both clinical and statistical)?*

We changed the sentence in line 207, Page 10 to better clarify how the meta-analysis will be performed. Regarding the studies heterogeneity, we intend to address the level of heterogeneity and discuss its interpretation depending on the findings. Possibly, it will include the discussion on how comparable the studies are and what limitations it can add to our findings/interpretations.

Comments from Reviewer 2

Nothing to add.

VERSION 3 – REVIEW

REVIEWER	Tormod Rogne NTNU, Norwegian University of Science and Technology Yale University
REVIEW RETURNED	27-Dec-2018
GENERAL COMMENTS	See previous comments. I am glad to see that sub-analysis stratified by when in pregnancy the test was taken has been included in the protocol. The specification of how the analyses will be carried out can still be more precise.

VERSION 3 – AUTHOR RESPONSE

Comments from Reviewer 2

- *I am glad to see that sub-analysis stratified by when in pregnancy the test was taken has been included in the protocol. The specification of how the analyses will be carried out can still be more precise.*

We tried to better clarify how we intend to conduct analyses as stated in the included sentences of lines 210-217 (Page 10). We also added two references that might support and elucidate our approach to the audience and other researchers. Therefore, we hope we have covered the required explanations on how our meta-analysis will be conducted. That said, we think we are not in condition to go further in this proposed analysis before having the review performed and the data already extracted.